# Human-in-the-loop Neural Networks: Human Knowledge Infusion

## Abstract

This study proposes a method for infusing human knowledge into neural networks. The primary objective of this study is to build a mechanism that allows neural networks to learn not only from data but also from humans. This motivation is triggered by the fact that human knowledge, experience, personal preferences, and other subjective characteristics are not necessarily easy to mathematically formulate as structured data, hindering them from being learned by neural networks. This study is made possible by a neural network model with a two-dimensional topological hidden representation, Restricted Radial Basis Function (rRBF) network. In rRBF, the hidden layer's low dimensionality allows humans to visualize the internal representation of the neural network and thus intuitively understand its characteristics. In this study, the topological layer is further utilized to allow humans to organize it considering their subjective similarities criterion for the inputs. Hence, the infusion of human knowledge occurs during this process, which initializes the rRBF. The subsequent learning process of rRBF ensures that the infused knowledge is inherited during and after the learning process, thus generating a unique neural network that benefits from human knowledge. This study contributes to the new field of human-in-the-loop (HITL) AI, which aims to allow humans to participate constructively in AI's learning process or decision-making and define a new human-AI relationship.

## 1 Introduction

In this study, we attempt to build a method for infusing human knowledge into neural networks. Recently, various topics of human-in-the-loop machine learning (HITL-ML) (Mosqueira-Rey et al., 2022; Xin et al., 2018) have been gaining much attention. The primary objective of HITL-ML is to allow humans to participate in the various stages of machine learning, for example, to curate the training data and arrange the training schedule, indicate a preferred learning direction, and participate in the decision-making process. Many attempts have been made to study how humans interact with AI in HITL-ML (Bengio et al., 2009; Dudley & Kristensson, 2018; Cui et al., 2021). The adoption of HITL in Reinforcement Learning has also gained attention (Christiano et al., 2017; Retzlaff et al., 2024), many of them include robot learning (Fitzgerald et al., 2018), where the infusion of human common sense prior to the RL process has been demonstrated to be beneficial (Ogawa & Hartono, 2022; Oriyama et al., 2024). HITL has also been applied to non-engineering fields (Chandler et al., 2022; Cohen et al., 2023).

In the past, human participation in machine learning was limited to curating training data, monitoring and changing the course of the training process, providing evaluation during the learning process, providing demonstrations, and taking active roles in decision-making. This study proposes a new way for human-AI interaction: intuitive infusion of human knowledge into neural networks. This is made possible by a layered neural network in this study, the Restricted Radial Basis Function (rRBF) network (Hartono et al., 2015; Hartono, 2020a). The rRBF has a two-dimensional hidden layer that is self-organized during the learning process to form label-relevant topological representations of high-dimensional inputs. The low dimensionality of the representations allows humans to visually understand the neural network; thus, so far, rRBF has been studied in the context of Explainable AI (XAI) (Hartono, 2020b; 2024). For this study, the two-dimensional hidden layer is utilized for infusing human knowledge in the initialization process of the RBF. Here, the low dimensionality of the hidden layer allows humans to hand-organize the high-dimensional input onto

it. This organization is based on the subjective knowledge and preferences of humans, so during the initialization process, knowledge infusion is executed. The infused knowledge is then inherited during and after the learning process, thus generating neural networks that bear the characteristics of their human initializers.

It should be noted that the objective of this research is not to develop neural networks that perform better than state-of-the-art models but to develop neural networks that can be seeded by human knowledge before learning from data. Naturally, the knowledge-infused plays an essential role in influencing the quality of knowledge-infused neural networks.

The proposed knowledge infusion mechanism is tested against Alzheimer's Disease data, demonstrating that it is possible to seed neural networks with human knowledge. The results open new vistas not only in HITL-ML but also in the co-relation of humans and AI.

## 2 HUMAN-IN-THE-LOOP LEARNING

The Human-in-the-loop (HITL) training in this study is made possible by using a hierarchical neural network with a 2-dimensional hidden layer, the Restricted Radial Basis Function (rRBF) network, introduced in (Hartono et al., 2015; Hartono, 2020a). The rRBF has a two-dimensional topological hidden layer, similar to that of Self-Organizing Maps (SOM) (Kohonen, 1982; 2013). The low-dimensional topological layer allows humans to visualize and thus intuitively understand the neural network's internal representation, which adds transparency to normally black-box hierarchical neural networks.

In this study, the two-dimensional internal layer is utilized not for visualization but for knowledge infusion. As this internal layer topologically represents the high-dimensional inputs, the distances between the internal representations reflect their similarities in their original high-dimensional space. Here, this topological characteristic is utilized for infusing human knowledge to the neural network. The idea is to allow a person (called an initializer) to hand-organize the internal topological layer, by assigning a small subset of the high-dimensional inputs into the two-dimensional hidden layer. The organization should be based on the initializer's subjective knowledge in which he/she assigns similar inputs close to each other while distancing dissimilar inputs. This internal organization becomes the initial state of the rRBF. The idea here is to allow humans to initialize the neural network before the training process, and thus infusing the initializer's knowledge or preferences into the neural network. Here, it is expected that the rRBF subsequently inherits the initializer's knowledge during and after the learning process.

We are aware that there are vast collections of dimensionality reduction methods that can be considered as the base for this study, for example, (van der Maaten & Hinton, 2008), (McInnes et al., 2018), (Zhang et al., 2018). However, we chose the rRBF's low-dimensionality due to its easy human interpretation. The chosen representation allows subjective and intuitive translation from high-dimensional labeled input similarities into two-dimensional relative distances on the representation space.

The dynamics and the knowledge infusion are elaborated in the subsequent sessions.

### 2.1 THE BASE NEURAL NETWORK

The base neural network, rRBF, is illustrated in Fig. 1.

At time $t$, observing input $\mathbf{x}(t) \in \mathbb{R}^d$, a best matching unit (BMU) among the hidden neurons is calculated as follows.

$$win(t) = \underset{j}{\operatorname{argmin}} \|\mathbf{w}_j(t) - \mathbf{x}(t)\| \tag{1}$$

Here, $\mathbf{w}_j \in \mathbb{R}^d$ is the reference vector associated with the $j$-th hidden neuron. The output of hidden neuron, $O_j^{hid}(t)$, is then calculated as follows.

$$O_j^{hid}(t) = \sigma(win(t), j)e^{-\|\mathbf{w}_j(t) - \mathbf{x}(t)\|^2} \tag{2}$$

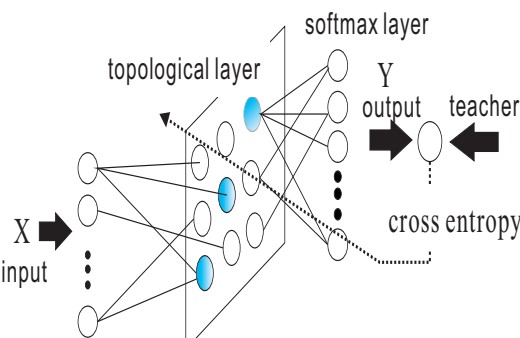

Figure 1: Base Network, rRBF, is a hierarchical supervised neural network with a two-dimensional topological hidden layer. In this hidden layer, neurons are arranged in a two-dimensional grid. The hidden neurons become two-dimensional representations of high-dimensional input. The hidden layer is fully connected to an output layer. In this study the rRBF is trained with cross-entropy loss function.

Here, $\sigma(win(t), j)$ is the neighborhood function, defined as follows.

$$\sigma(win, j, t) = e^{-\frac{\|pos(win(t)) - pos(j)\|^2}{S(t)}} \tag{3}$$

Here, $pos(j) \in \mathbb{R}^2$ is the two-dimensional coordinate of the $j$-th neuron on the topological hidden layer, while $S(t)$ is the annealing function defined as follows.

$$S(t) = N_\infty + \frac{1}{2}(N_0 - N_\infty)(1 + cos(\frac{\pi t}{t_\infty})), \tag{4}$$

where $N_0 > N_\infty > 0$ are emperically decided constants, and $t_\infty$ is the termination iteration for the learning process.

The output of the rRBF, $\mathbf{O(t)} \in \mathbb{R}^{N_c}$, where $N_c$ is the number of class labels, is then calculated as follows.

$$\mathbf{O}(t) = softmax(\mathbf{V}(t)\mathbf{O}^{hid}(t)) \tag{5}$$

Here, $\mathbf{V}(t) \in \mathbf{R}^{N_c \times N_H}$ is the weight matrix between the hidden and the softmax layer, while $\mathbf{O}^{hid}(t) = \begin{pmatrix} O_1^{hid}(t) \\ \vdots \\ O_{N_H}^{hid}(t) \end{pmatrix}$, and $N_H$ is the number of the hidden neurons.

Setting a loss function, $L(\mathbf{O}(t), \mathbf{T}(t)$, where $\mathbf{T}(t)$ is the one-hot-encoded teacher signal for input $\mathbf{x}(t)$, the rRBF can be trained, by calculating the derivatives $\frac{\partial L}{\partial \mathbf{V}(t)}$, and $\frac{\partial L}{\partial \mathbf{w}_j(t)}$.

In this study, cross-entropy is utilized as the loss function.

In CRSOM, the modification to the reference vector $\mathbf{w}_j$ at time $t$ is as follows.

$$\begin{aligned} \Delta\mathbf{w}_j(t) &\propto \frac{\partial L}{\partial \mathbf{w}_j} \\ &\propto \delta_j(t)O_j^{hid}(t)(\mathbf{w}_j(t) - \mathbf{x}(t)) \end{aligned} \tag{6}$$

Here, $\delta_j(t)$ is the regularization signal from the output layer to the $j$-th hidden layer, as follows.

$$\delta_j(t) = v_{jK}(t) - \sum_l v_{jl}(t)O_l(t) \tag{7}$$

In Eq.7, $v_{jl}$ is the connection weight from the $j$-th hidden neuron to the $l$-th neuron in the softmax layer, , $O_l$ is the value of the $l$-th neuron in the softmax layer at time $t$, while $K$ is the ground truth class of the input $\mathbf{x}(t)$ assuming one-hot encoding for the teacher signal. It is evident here that this regularization signal is affected by the ground truth of the input.

This regularization signal distinguishes the modification of the reference vector in rRBF from that of SOM. If $\delta_j > 0$, then the modification is identical to SOM. However, here $\delta_j$ can be positive or negative. Hence, while in SOM, the modification is not affected by the label of the input, the label plays an essential role in deciding the direction of the modification (Hartono, 2020a), consequently the hidden layer is called Context-Relevant Self-Organizing Maps (CRSOM). The detail of the derivation of this modification is given in the Appendix B

In this study, the infusion of human knowledge is executed by hand-organizing the CRSOM before the learning process.

## 2.2 HITL Pre-training

For infusing knowledge, a person (an intializer) is asked to map and organize a small subset of the inputs on the 2-dimensional hidden layer of the rRBF.

Let $Z = \{\mathbf{z}^1, \mathbf{z}^2, \cdots, \mathbf{z}^L\} \subset X$ be the instances to be organized by human initializer, where $X = \{\mathbf{x}^1, \mathbf{x}^2, \cdots, \mathbf{x}^N\}$ is the set of all instances, and $L$ is the number of instances that are available for human initializer to organize while $N > L$ is the total number of instances. Let $D_{ij}$ be a weighted distance between the $i$ and the $j$ samples defined as follows.

$$D_{ij} = \|\Lambda^t(\mathbf{z}^i - \mathbf{z}^j)\|^2 \tag{8}$$

Here $\Lambda = (\alpha_1, \alpha_2, \cdots, \alpha_d)^t$ is an attention vector that weighs the importance of the elements of the input.

Let $d_{ij}$ be the distance between the representation of the $i$-th and the $j$-th inputs arranged by the human initializer on the hidden representation defined as follows.

$$d_{ij} = \|pos(i) - pos(j)\|^2 \tag{9}$$

Here $pos(i) \in \mathbb{R}^2 (i = 1, 2, \cdots L)$

A loss function $L_\Lambda$ that measures the discrepancies between the initializer's hand arrangements and the weighted distances between two inputs is defined as follows.

$$L_\Lambda = \sum_{1 < i < j < L} (D_{ij} - d_{ij})^2 \tag{10}$$

Minimizing $L_\Lambda$ with gradient descent as follows

$$\Lambda(t + 1) = \Lambda(t) - \eta \frac{\partial L_A}{\partial \Lambda(t)} \tag{11}$$

The resulting attention vector is $\Lambda^*$. As the attention vector reflects the initializer's hand-organized representation, it is the quantitative representation of the initializer's subjective knowledge. The infusion of human's knowledge is then executed through this attention vector as follows. It is possible to execute more complicated dimensionality reductional methods, for example, (Goldberger et al., 2004) or (Tenenbaum et al., 2000). However, for the simplicity of interpretation of the extracted metrics and humans' intuitiveness in translating input differences into internal representation differences, we utilized simple Multi-Dimensional Scaling (MDS).

## 2.3 Training and Re-training

The training process of the rRBF is executed as explained in subsection 2.1, except that the input $\mathbf{x}$ is substituted with $(\Lambda^*)^t\mathbf{x}$. This is where human knowledge plays an essential role during learning.

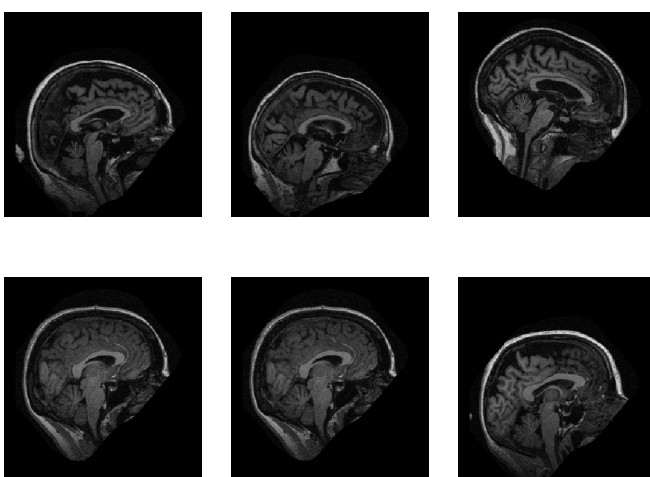

Figure 2: Samples from OASIS-1. The data was a collection of 836 instances of $256 \times 256$ pixels, acquired from 235 subjects aged 33 to 96. These data were made available by the OASIS Brains Project of Washington University School of Medicine in St. Louis.

The initializer's knowledge now filters the input. Consequently, the learning of rRBF produces a topological representation developed from human-initialized representations. The CRSOM further allows humans to make some corrections by modifying the representation in the same manner as in the pre-training stage. The correction is made, for example, to improve the neural network's performance.

An example of human initialization and the correction process is explained in the next section.

## 3  EXPERIMENTS

In the experiment, cross-sectional brain MRI data for mild to moderate Alzheimer's Disease (AD) detection provided by Open Access Series of Imaging Studies (Marcus et al., 2007) were used. Some samples are shown in Fig. 2. The task is a binary classification of normal or AD for each input.

From the data, 40 images were taken to be hand-organized by human initializers. Each initializer was asked to organize the images according to his/her preferences on the topological layer of the rRBF. Six initializers participated in this experiment; none were medical experts, so the images were organized based on their appearances and labels. Each initializer made an educated guess that similar images with identical labels should be assigned close to each other, and images with different appearances should be distanced. We understand that this assumption may differ from medical insight, but it is enough to infuse common-sensical knowledge into neural networks at this stage. After the HITL initialization, the rRBF was trained with training data of 637 and then tested against testing data of 159.

Hereafter, human-initialized rRBF will be denoted as HITL-rRBF.

Figure 3 shows three initial organizations arranged by three different initializers. It can be observed that they have distinctive topological organizations that reflect the difference in their common sense. The difference in the heatmaps depicts the difference in each initializer's common sense. It can be observed that the resulting CRSOMs inherit the human-organized initial representations. This demonstrates that it is possible to infuse human knowledge into neural networks.

The HITL process is not necessarily executed only at the start of the learning process but at any stage. Here, after the termination of the learning process, humans can visualize the CRSOM and find some regions where the HITL-rRBFs are likely to misclassify. This can be done by finding tangled representations, such as adjacent neurons on the CRSOM with conflicting labels. The correction process is shown in Fig. 4. It can be observed from this figure that once humans correct

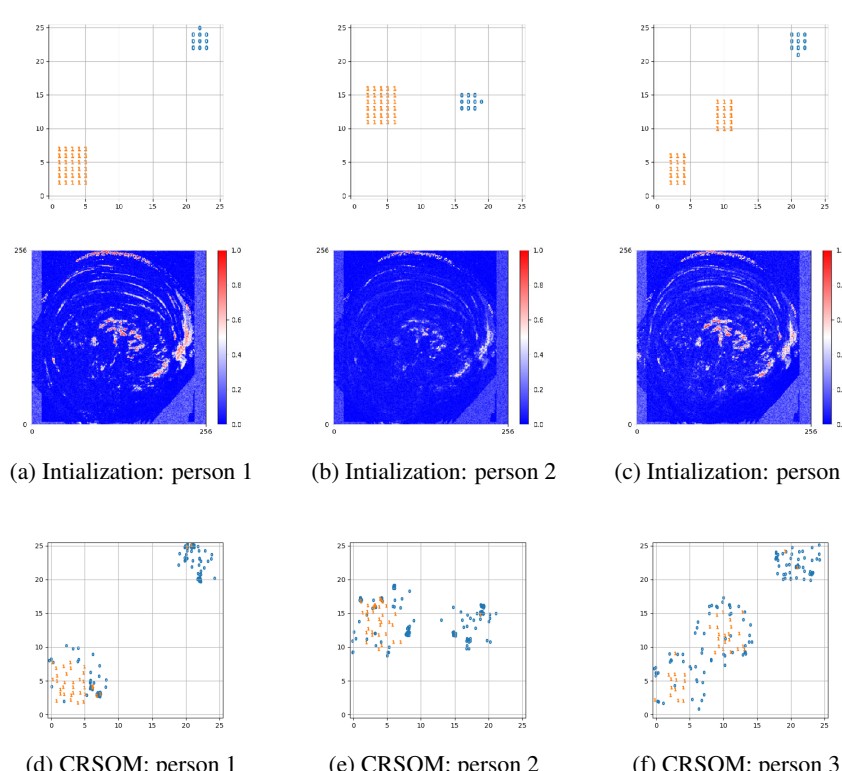

(a) Intialization: person 1      (b) Intialization: person 2      (c) Intialization: person 3

(d) CRSOM: person 1      (e) CRSOM: person 2      (f) CRSOM: person 3

Figure 3: Results of HITL-rRBF. The first row of (a)-(c) shows the initial maps by each initializer (Different colors in the maps denote different labels), while the second row shows the resulting attention vectors in Eq.11, expressed as heatmaps. (d)-(f) show the resulting CRSOMs.

the initial organization, heatmaps are also corrected. While the resulting CRSOMs still inherit the initial organizations, they develop more disentangled maps reflecting humans' correction.

The accuracy of the rRBFs over the test data is shown in Fig. 5 (a). It can be observed that most HITL rRBFs perform better than the randomly initialized rRBF. This is understandable as the initialization usually provides an initial internal representation that is common-sensically better organized than random initialization. Figure 5 (b) shows that HITL-rRBF can learn smoothly.

The experiments also show that human initializations produce diversity in the neural networks as reflected by their different performance in Fig. 5 (a). This is the most critical aspect of this study, in which the main objective is to build neural networks that learn from data and humans. In this light, it is natural that the performance of the neural networks also depends on the knowledge of their human initializers. The rRBFs benefit from the quality of the knowledge infused into them; in this case, some initial knowledge produces rRBFs that are better than CNN.

For comparisons, the HITL-rRBF is compared to the randomly initialized rRBFs. (a)-(c) in Fig. 6 shows three examples of non-HITL-rRBF that were randomly initialized. It is obvious from Fig. 3, Fig. 4, and this figure that the HITL-rRBF produced more label-organized maps than the randomly initialized ones. Furthermore, t-SNE (van der Maaten & Hinton, 2008) and UMAP (McInnes et al., 2018) representations illustrate the natural data distribution of this problem. The difference between the CRSOMs in Fig. 3 and the non-HITL maps in Fig. 6 indicate that human knowledge infusions substantially change the representation of the neural networks in dealing with the given problem. In the next experiment, the HITL-rRBF was against the handwritten digit data, MNIST. This experiment aims to test the influence of the quality of the infused knowledge on the rRBF's learning ability. While for humans, the similarities between the samples of MNIST data are also subjective; they are more sensible in that humans can easily explain the reasons behind their hand-organized

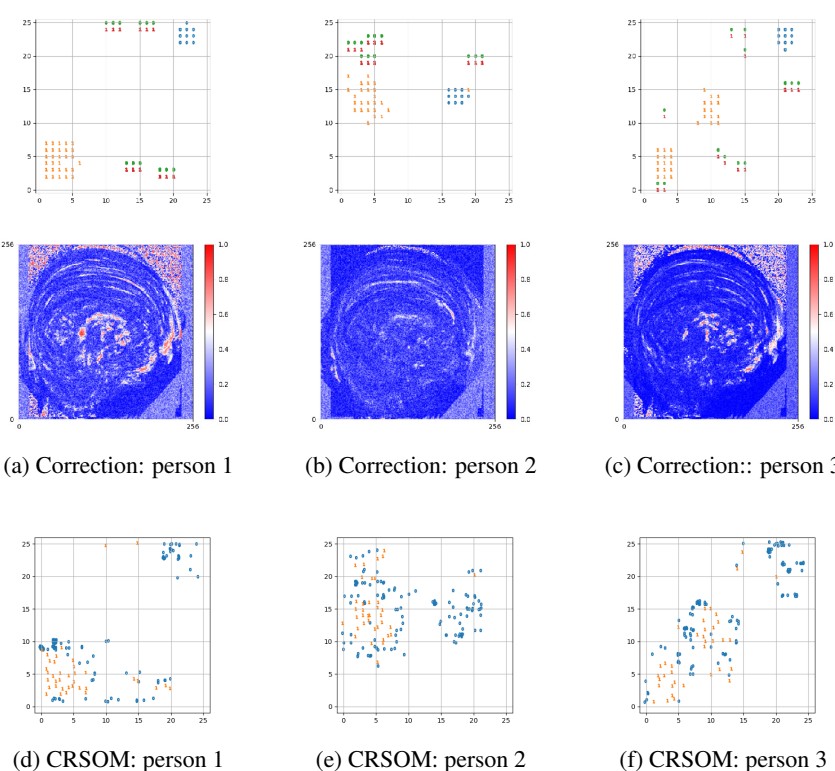

(a) Correction: person 1    (b) Correction: person 2    (c) Correction:: person 3

(d) CRSOM: person 1    (e) CRSOM: person 2    (f) CRSOM: person 3

Figure 4: Corrected HITL-rRBF. This figure shows the human-corrected maps, the resulting heatmaps and the post-correction CRSOM in the same configuration as Fig.3.

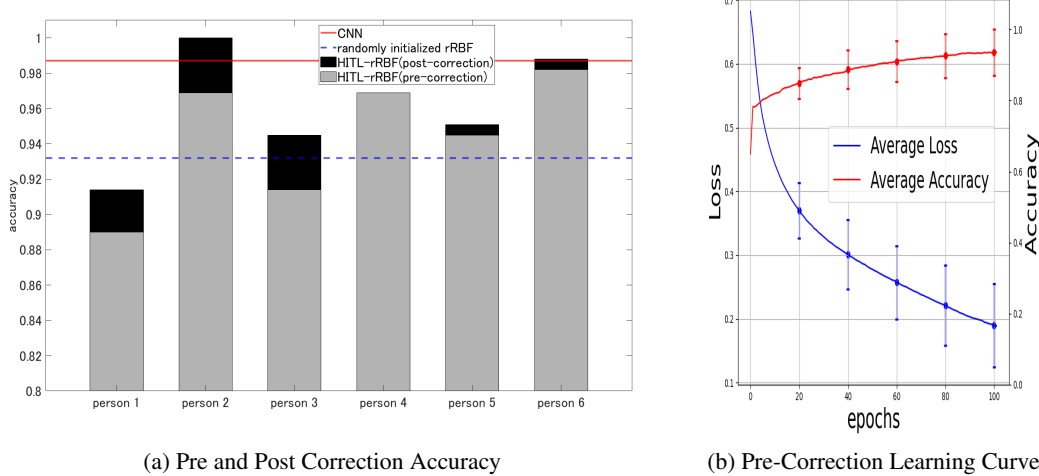

(a) Pre and Post Correction Accuracy    (b) Pre-Correction Learning Curve

Figure 5: Pre and Post Correction Accuracy. (a) shows the generalization accuracy of each initializer's HITL-rRBF after the termination of its learning process (grey) and the improvement after human correction of its internal representation (black). The blue cut-off line denotes the the average accuracy of some non-HITL rRBF randomly initialized, while the red cut-off line is the average accuracy of CNNs(three convolutional layers, each one followed by a pooling layer, and subsequently two fully connected layers and finally a softmax layer). Each CNN has an identical structure but intialized differently. (b) shows the learning curve averaged over six HITL-rRBF.

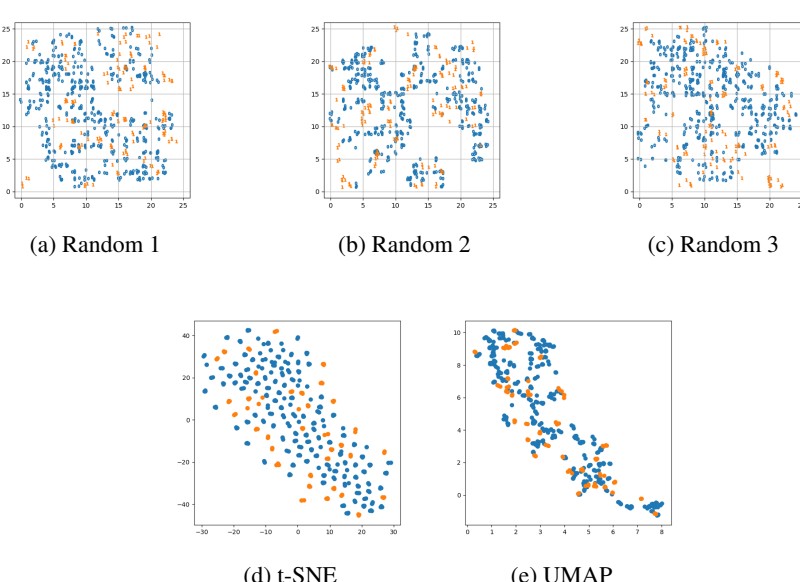

(a) Random 1      (b) Random 2      (c) Random 3

(d) t-SNE      (e) UMAP

Figure 6: non-HITL CRSOMs. (a)-(c) are the three samples of CRSOMs of non-HITL rRBFs that were randomly initialized without human knowledge infusion. At the same time (d) and (e) are the two-dimensional representations of this problem by t-SNE and UMAP.

initial maps. For example, most humans think that "1" and "7" are similar, while "3" and "4" are dissimilar.

The initial organizations and their resulting CRSOMs are shown in Fig. 7. Here, humans' sensible initializations and deliberately made non-sensical initial maps with their resulting CRSOMs are shown. The resulting CRSOMS shows that the non-sensical initializations tend to produce truncated clusters belonging to the same labels and form mixed clusters of conflicting labels that may result in misclassification. This experiment is executed to observe the effect of different expertise levels infusion into the rRBF, where non-sensical initializations emulate poor expertise. The results are shown in Fig. 8. Our objective is not to build a new neural network that outperforms SOTA models but to propose a novel idea for infusing human knowledge into neural networks. The empirical results show that the neural network inherits the expertise level of its human initializer, confirming the viability of our idea.

Figure 8 shows that the quality of the infused knowledge matters to the learning results of the HITL-rRBF. Here it is apparent that the infusion of the poor knowledge translates to a slower learning process and lower generalization abilities of the HITL-rRBF.

In this study, experiments involving human initializers were conducted following the ethical guidelines released by the authors' institution. The participants were engineering students aged 21-24 without expertise in medical science. They understood the objective of the knowledge infusion experiment but were not any part of this study and did not have a deep understanding of machine learning.

## 4 CONCLUSION

In this study, we proposed an efficient way to infuse human knowledge into neural networks. This is made possible by the structure of the hidden layer of rRBF. Our primary motivation here is to extend the learning ability of neural networks that normally learn from data or from their interaction with their environments, in the case of Reinforcement Learning, but not directly from human knowledge, preferences, or experiences. The subjectivity of human knowledge and preferences is often difficult to formulate mathematically and express as structured data and, thus, difficult to be presented to neu-

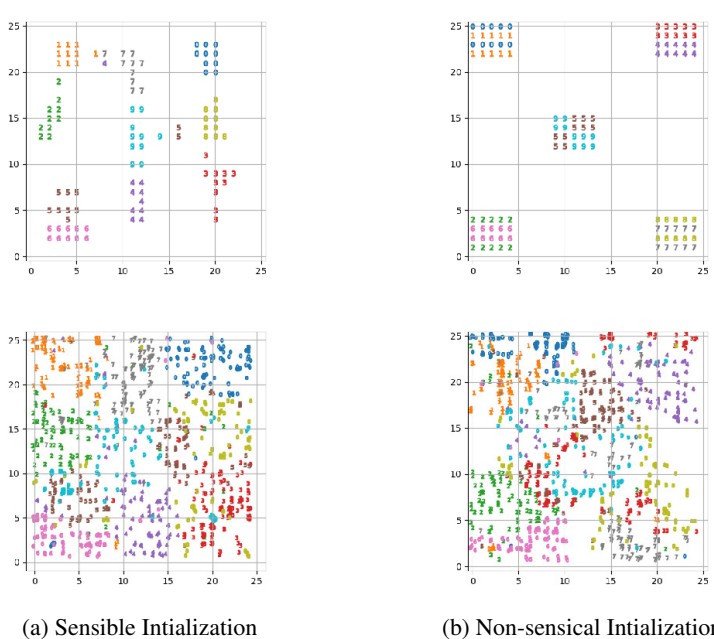

(a) Sensible Intialization  (b) Non-sensical Intialization

Figure 7: Experiment on MNIST. The first row of (a) shows a sample for sensible initialization by humans, for example, by placing "1"s and "7"s closely while distancing "6" from them, while the second row of (a) shows the resulting CRSOM. The first row of (b) shows the deliberate non-sensical initialization, for example, by mixing some "6"s with "2"s, "3"s with "4"s, etc, while the second row shows the resulting CRSOMs.

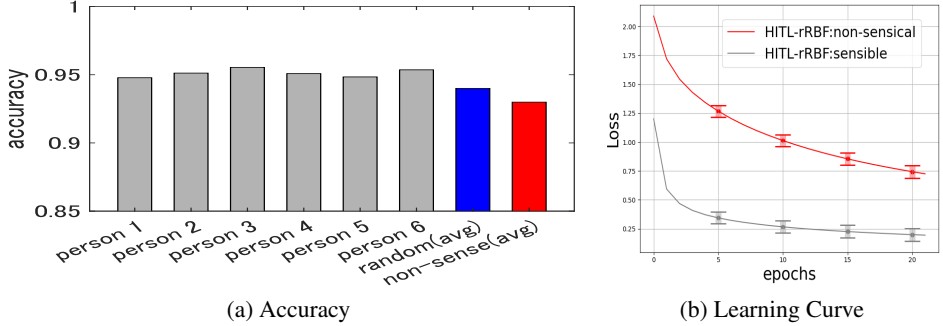

(a) Accuracy  (b) Learning Curve

Figure 8: Accuracy and learning ability of different expertise infusions. (a) shows the generalization accuracies of six different HITL-rRBFs that were sensibly initialized (grey bars) and the average performance of non-HITL-rRBFs that were randomly initialized (blue bar) and the average performance of HITL-rRBF infused with poor expertise (red bars). (b) shows the average learning process of HITL-rRBF. The red curve is for the non-sensically initialized rRBF while the grey curve is for the sensibly initialized rRBF.

ral networks to learn. This study attempts to fill the gap by allowing humans to intuitively organize a neural network's representations and infuse their knowledge during the organization process.

It should be noted that the base neural network, rRBF, is essential for this objective. Here, it is important for humans to understand the relations of high-dimensional inputs' representation in the base neural network. In rRBF humans can subjectively or common-sensically project high-dimensional inputs onto the two-dimensional hidden layers, because the relative two-dimensional positions on the hidden layer can be subjectively translated into similarities of the high-dimensional inputs. This translation from inputs to hidden representation cannot be done on most layered neural networks as humans can't understand the hidden representations of the neural networks.

We are aware that this study is still in its preliminary stage. However, the experiments show promising results in demonstrating that infusing human knowledge into neural networks is possible. For this research, the knowledge levels of the human initializers are indistinguishable. Hence, linking the resulting neural networks' expertise level with that of their human initializers is impossible. This will be our immediate future work.

The ability of neural networks from humans will form a new relationship between humans and AI. It will be possible to use personalized AI that can better emulate humans. It will also be possible to use AI as a kind of knowledge preservation mechanism that can subsequently be used for knowledge transfer: not only transfer from humans to machines, but also machines to humans, and most interestingly, human to human using AI as an intermediary.

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

## A  APPENDIX 1

To ensure the reproducibility of this study, we submit the source code of the simulation in Python, RUNTHIS.ipynb. The text file version is given in code_in_text.txt. This simulation assumes that the human initialization has been given. The simulation then runs the extraction of the attention vector and the learning process of the rRBF. At the end of the simulation, the heatmap of the attention vector and the CRSOM will be produced. Depending on the computer's power, the simulation may need considerable time to run. The training data size was reduced by 10% to accommodate the file size limitation. Due to the smaller size of the training data, the rRBF produces slightly different results from the ones in the paper.

## B  APPENDIX 2: CRSOM'S MODIFICATION DERIVATION

Here, the derivation of the reference vector modification in CRSOM in Eq. 6 is elaborated.

Let $O_k(t)$ be the value of the $k$-t neuron in the softmax layer in Eq.5. The loss function, $L(\mathbf{x}(t))$,for the input at time $t$ is as follows.

$$L(\mathbf{x}(t)) = -\sum_k T_k log O_k(t) \tag{12}$$

When $T_k(t)$ is the $k$ component of the one-hot encoded teacher signal ($T_K(t) = 1$ when $k = K$ is the ground truth label for input $\mathbf{x}(t)$, and $T_k(t) = 0$ otherwise).

Because $O_k(t) = \frac{e^{I_k(t)}}{\sum_l e^{I_l(t)}}$, for $I_k(t) = \sum_j v_{jk} O_j^{hid}(t)$ where $v_{jk}$ is the connection weight from the $j$-th hidden neuron to the $k$-th neuron in the softmax layer, the modification can be calculated as follows.

$$
\begin{aligned}
\frac{\partial L(t)}{\partial \mathbf{w}_j(t)} &= \frac{\partial I_K(t)}{\partial \mathbf{w}_j(t)} - \frac{\partial (log(\sum_l e^{I_l(t)}))}{\partial \mathbf{w}_j(t)} \\
&= \left( \frac{\partial I_K(t)}{\partial O_j^{hid}(t)} - \frac{\partial (log(\sum_l e^{I_l(t)}))}{\partial O_j^{hid}(t)} \right) \frac{\partial O_j^{hid}(t)}{\partial \mathbf{w}_j(t)} \\
&= \left( v_{jK} - \sum_l \frac{\partial e^{I_l(t)}}{\partial O_j^{hid}(t)} \frac{1}{\sum_l e^{I_l(t)}} \right) \frac{\partial O_j^{hid}(t)}{\partial \mathbf{w}_j(t)} \\
&= -2 (v_{jK} - \sum_l v_{jl} O_l(t)) \, O_j^{hid}(t) \, (\mathbf{w}_j(t) - \mathbf{x}(t))
\end{aligned}
\tag{13}
$$

Hence for $\delta_j(t) = (v_{jK} - \sum_l v_{jl} O_l(t))$, the modification $\Delta \mathbf{w}_j(t) \propto \delta_j(t) \, O_j^{hid}(t) \, (\mathbf{w}_j(t) - \mathbf{x}(t))$

Here, as $O_j^{hid}(t) > 0$, it should be noted that for $\delta_j(t) > 0$, the reference modification for CRSOM is proportional to the reference vector modification in standard SOM, $\Delta \mathbf{w}_j^{SOM}$. However, as $\delta_j(t) \in \mathbb{R}$, $\Delta \mathbf{w}_j \neq \Delta \mathbf{w}_j^{SOM}$, and thus CRSOM and SOM produce substantially different topological maps.

