# OpenReview forum: "Human-in-the-loop Neural Networks: Human Knowledge Infusion"
_ICLR.cc/2025/Conference — Submitted to ICLR 2025_

### Official Review · Reviewer_gtmY · 2024-10-30

**Soundness:** 1
**Presentation:** 2
**Contribution:** 1
**Rating:** 3
**Confidence:** 4

**Summary:**

This paper proposes a new method for infusing human knowledge into neural networks. It builds upon the Restricted Radial Basis Function (rRBF) network, similar to Self-Organizing Maps,   to infuse the knowledge by initializing the input based on human preferences. The proposed algorithm is evaluated on a brain MRI dataset for Alzheimer’s diagnosis.

**Strengths:**

• Infusing human knowledge in neural networks is a relevant topic , yet most studies focus on reinforcement learning, leaving other knowledge distillation techniques under-explored;

• The idea of mapping the input data to a new representation space respecting human preferences is interesting and novel;

• Alzheimer’s disease detection from brain MRI is still a challenge, especially in the early stage.

**Weaknesses:**

• The main idea of this paper – initializing a neural network using a new data representation based on human preferences – is unrelated to the specific network architecture. The authors focused this study on a single architecture (rRBF), arguing for better interpretability. Yet, all the results presented could have been generated using any kind of deep neural network (CNN, Transformer, MLP, etc.). For instance, Fig. 4 could have been generated using the internal representation of a network at various depths. As reported in Fig. 7, the performance of rRBF is quite low compared to a simple CNN, which could have been expected by its shallow architecture (2 layers). I do not understand why the authors made such a choice and I think it highly limits the current experimental setup used to validate the method. Showing the benefit of their method on different families of DNN would highly improve the experimental design.

• My second concern, as expected by the authors, is about the pool of human initializers used to judge image similarities. They are not medical doctors, and their personal opinion about the similarity between two brain MRIs is highly questionable. This is easily seen in Fig. 7 where the model’s performance is no better than a simple CNN when using human judgment for 5 over 6 individuals. I recommend using a pool of medical doctors to perform this task. In this case, it should be interesting to understand the inter-individual differences between image similarities as judged by this pool of doctors.

• The authors only performed experiments on a small brain MRI dataset (235 subjects), although they claim a very broad method. Additionally, no statistical tests or cross-validation schemes were performed to evaluate and compare the models (e.g., in Fig. 7). I would first recommend using a much larger dataset (such as ADNI for Alzheimer’s disease) and studying harder tasks (e.g., diagnosing MCI vs AD vs Controls) to clearly show the benefit of using human knowledge in a real-life scenario.

• Section 2.1 (describing the rRBF architecture) is unclear and I had to read the original papers from (Hartono, 2015, 2020) to clearly understand all the technical details. Besides, as I mentioned previously, I think the exact architecture is irrelevant in the proposed method and it does not add novelty to the current work (e.g. Fig. 1 is not novel per se as it only describes an rRBF network). I would recommend shortening this section, moving technical details to the appendix and re-focusing on the actual novelty of this work (which is the human infusion technique in Section 2.2).

**Questions:**

• Related to my 1st point in the weaknesses, why did the authors choose specifically rRBF networks in this work over more classical networks (CNN…) ?

• In section 2.2, you mentioned that you solved a linear Multidimensional Scaling (MDS) problem to map the input data to a new representation space. Did you consider non-linear MDS techniques (IsoMap [1], Laplacian Eigenmaps [2], etc…) ?

•  In Fig.3, you show attention maps on a brain MRI that you obtained by solving the linear MDS problem on human judgments. They seem hard to interpret as very different areas are highlighted (frontal lobe, ventricles, etc…). Did you perform a statistical analysis to retain only the significant regions? A finer analysis would be interesting to compare the inter-individual differences between human annotators.

[1] A global geometric framework for nonlinear dimensionality reduction, Tenenbaum et al., Science 2000
[2] Belkin, M., & Niyogi, P. (2003). Laplacian eigenmaps for dimensionality reduction and data representation, Belkin et al., Neural Computation, 2003

---

> ### Author Response · Authors · 2024-11-25
> **response to reviewer's questions (Nov. 25)**
>
> We thank you very much for your insightful comments and criticisms.
> Our point-to-point response is as follows:
>
> Q1:  Related to my 1st point in the weaknesses, why did the authors choose specifically rRBF networks in this work over more classical networks (CNN…) ?
>
> R1: Thank you for raising this important issue. Contrary to your comment, rRBF in this study is essential for implementing the knowledge infusion and cannot be replaced by another network like CNN. The reason is that the two-dimensional topological hidden layer of the rRBF offers us an intuitive understanding of the hidden representations, i.e., similar high-dimensional inputs are positioned close to each other, while dissimilar inputs are distanced on the map. This understanding allows humans to infuse their subjective knowledge by making topological arrangements on the map. In short, the rRBF offers direct translation for inputs's similarity on their original high-dimensional space into their relative distance on the low-dimensional representation space of the rRBF. In CNN, human cannot directly translate the difference of the input in their original high-dimensional space into the difference in their hidden representation. This explanation is added in lines 489-494.
>
> Q2:  In section 2.2, you mentioned that you solved a linear Multidimensional Scaling (MDS) problem to map the input data to a new representation space. Did you consider non-linear MDS techniques (IsoMap [1], Laplacian Eigenmaps [2], etc…) ?
>
> R2: It is possible to consider non-linear MDS however in this research we deliberately chose the standard linear MDS. The primary reason is interpretability, in that it is substantially easier for humans to translate the perceived dissimilarity of two different inputs with two-dimensional differences in the representation space. The non-linearity of IsoMap or Laplacian Eigenmaps will make this task difficult for human initializers.
>
> Q3: A finer analysis would be interesting to compare the inter-individual differences between human annotators.
>
> R3: This is also an important point. We admit that, at this point, we cannot perform a deeper analysis of inter-individual differences for this problem. However, we ran additional experiments using MNIST to analyze the differences between good initialization and the poor one. While MNIST is a simple problem, it allows us to execute additional experiments to test the proposed idea against the variety and quality of the infused knowledge. While humans subjectively perceive samples of MNIST, they can give a clear rationale for their perception. For example, most humans will consider digits "1" and "7" to be similar and digits "4" and "8" to be dissimilar. This will give variety on the subjectivity of the infused knowledge, including non-sensical knowledge infusion, i.e., organizing dissimilar digits close to each other. In the new experiments, we show that the quality of the infused knowledge subsequently influences the neural network, strengthening our argument that it is possible to build a neural network that learns from humans. This explanation is added in lines 321-404 and 436-461.

---

> > ### Comment · Reviewer_gtmY · 2024-11-26
> > **Response to the Authors**
> >
> > Thank you for the clarifications. However, my concerns about the model's evaluation (only limited to a tiny brain MRI dataset) still stand. Moreover, it is still not clear why non-linear MDS should lack interpretability since the construction of the distance matrix (d_ij) by the human initializers is unrelated to the matrix (D_ij). To me, linear MDS limits the broadness of the proposed method as using a simple linear filter would be arguably inefficient for complex natural images.
> >
> > Minor: I still do not see the explanations provided by the Authors l.264-269.

---

> ### Author Response · Authors · 2024-11-25
> **Responses on the comments about the weakness of the proposed work (Nov. 25)**
>
> Thank you very much for your insightful comments on the weakness of our proposed model.
>
> Comment: I recommend using a pool of medical doctors to perform this task. In this case, it should be interesting to understand the inter-individual differences between image similarities as judged by this pool of doctors.
>
> Response: Yes, testing the mechanism using a pool of medical doctors is necessary. Unfortunately, as experiments using doctors are difficult to execute, we cannot perform them at this point. At this stage, we want to establish a solid method that will allow us to test our proposed HITL mechanism further for real-world problems. To complement this weakness, we ran tests against MNIST that, to some extent, allowed us to observe the effect of the different levels of expertise.
>
> Comment: Section 2.1 (describing the rRBF architecture) is unclear and I had to read the original papers from (Hartono, 2015, 2020) to clearly understand all the technical details.
>
> Response: As the rRBF is not a standard neural network, we think it will be good for broad readers to have brief explanation on the rRBF. Furthermore, the original rRBFs in [Hartono 2015, 2020] were trained using the squared error loss function, while the rRBF here was trained with cross-entropy and thus produced different modification rules.  We add an explanation of the modification rule in lines 152-161 and elaborate in Appendix B in lines 612-641.

---

> ### Author Response · Authors · 2024-11-26
> **Response to reviewer's comment**
>
> I sincerely appreciate your time and effort in thoroughly checking our revised paper and providing further discussion.
>
> We admit that we cannot resolve the problems of our limited evaluation during this two-week revision period. After establishing a solid framework for deeper evaluation, it is our immediate future task.
>
> As for the non-linear MDS, I apologize for the unclarity in my previous response. We assume that it is more intuitive for humans to interpret the difference of high-dimensional inputs into their simple Euclidean distance rather than kernel-based distance. Human intuitiveness in infusing knowledge is essential here. It is also related to implementation simplicity in that linear MDS directly produces an attention vector, as mentioned in line 204, that can be directly utilized for the inputs to the rRBF. I agree with the reviewer that we need to try different means of MDS to improve the performance of the proposed idea further.
>
> I'm sorry for my last mistake in writing the lines. It should be line 208-211.
>
> Once again, I thank you very much for initializing an insightful discussion.

---

### Official Review · Reviewer_PDph · 2024-10-31

**Soundness:** 3
**Presentation:** 2
**Contribution:** 2
**Rating:** 5
**Confidence:** 4

**Summary:**

This paper proposes a method for directly infusing human knowledge into data-driven neural networks, based on the rRBF network, and attempts experimental validation. The authors named the hidden layer where human knowledge is infused as the Context-Relevant Self-Organizing Maps (CRSOM). This process can be executed both at the early or intermediate stages of the network's learning process. Experiments were conducted using an Alzheimer’s MRI dataset with six initializers participating. The results indicate that networks infused with human knowledge via the proposed method show the potential for superior performance compared to the baseline.

**Strengths:**

1. Infusing human knowledge into machine learning systems (or vice versa) remains an unresolved topic in the field of human-in-the-loop (HITL), and various approaches to address this challenge should be encouraged. This paper presents an attempt that can be positively evaluated in this regard. Although the study is limited to a classification problem, it could potentially be extended to broader applications, including reinforcement learning.

2. The mathematical formulation and flow in Sections 2.1 and 2.2 are relatively clear, which may benefit readers with diverse backgrounds.

3. Conducting experiments using complex, real-world medical imaging data is a reasonable approach, as it demonstrates the robustness of the proposed method.

4. The paper acknowledges its limitations explicitly in the introduction.

5. Despite the various weaknesses mentioned below, I believe this paper has considerable potential to be improved and developed more robustly in the future.

**Weaknesses:**

1. The authors state that the aim of this study is not to develop state-of-the-art models (Line 64). To acknowledge the contribution of this study, however, it is necessary to introduce a novel concept (an innovative methodology or rigorous human behavioral experiment results). These contributions, however, appear somewhat lacking. For example, the learning process of a network similar to SOM is known as a clustering process that corresponds to the high-dimensional space. If CRSOM identifies clusters that align with the arbitrary classified samples by the initializer, the authors' methodology could simply be considered a variant of SOM adapted to human prior knowledge (i.e., an application case).

2. As I understand only six subjects participated in the experiment. Given that the subjects were laypersons rather than experts, it is feasible to recruit more participants. However, with only six participants, it is difficult to ascertain the statistical significance of the experimental results.

3. For an experiment involving human subjects, it is necessary to describe the recruitment process, the participants’ characteristics, and whether IRB approval was required for the study, which is currently missing.

4. Even if it is accepted that expert involvement is not necessary at this stage of study, if non-experts evaluated MRI images for similarity, this judgment might not be substantially different from the similarity that an unsupervised learning model, such as an autoencoder, could learn. What if the autoencoder had instead learned and provided similarity information for these images that were then infused into CRSOM, rather than using human initializers? Given the domain of the experimental images, the general knowledge of non-experts could be within the range that the model could deduce independently. Thus, even if knowledge infusion is feasible, further examination may be needed to confirm whether the information infused was indeed uniquely human (i.e., unobtainable by the model itself).

5. There are several inaccuracies or omissions in the presentation. For instance, the figure legends tend to be insufficient. What does CROM refer to in Fig. 4? Is it CRSOM? Even if so, the explanation remains somewhat unclear. Personally, I suggest condensing Fig. 2 and Fig. 7 as they are somewhat disproportionate in size relative to the key information. Instead complementing the text to provide more detail on the human experimental procedures may be recommanded.

**Questions:**

1. The terms "re-learning" and "re-training" appear multiple times throughout the text. Do they have the same meaning? If so, is there a reason to differentiate them?

2. Although the objective of this study is not necessarily to propose a high-performance model, from a practical perspective, the proposed methodology underperforms compared to CNN. Would it not be more beneficial to integrate CRSOM into CNN and compare this with a baseline CNN instead?

3. In the main text, should Fig. 6 on line 315 be corrected to Fig. 5?

4. Overall, the figure legends are insufficient. Should the legend in Fig. 4 refer to CRSOM rather than CROM?

5. The legend in Fig. 7 lacks clarity. For example, a clearer term such as "rRBFs before the human corrections were made" could replace "learning."

6. While a "standard CNN" is mentioned, could you specify what is meant by a "standard CNN"?

7. With only six participants, does this study have sufficient statistical power?

8. Has ethical consideration been given to the use of human subjects in this experiment, including IRB approval?

9. In Fig. 3, what does "upper" precisely refer to? Does it represent a cognitive similarity metric as perceived by the initializer? If my understanding is correct, would it not be somewhat unnatural for perceived human similarities to appear as uniformly regular grid-like arrangements?

**Details Of Ethics Concerns:**

This study involves human behavioral experiments, yet there is no clear mention of IRB approval.

---

> ### Author Response · Authors · 2024-11-25
> **response to reviewer Reviewer PDph (Nov. 25)**
>
> We want to thank you very much for your insightful comments and criticisms on the weakness of our paper.
> We have now revised our paper to improve the technical and writing quality. We hope that the revised paper is now acceptable for this conference.
>
> Here are our point-to-point responses to the reviewer's comments.
>
> Q1: The terms "re-learning" and "re-training" appear multiple times throughout the text. Do they have the same meaning? If so, is there a reason to differentiate them?
>
> R1: Those two terms have the same meaning. In the revised version, we have use the term "re-training" throughout the paper.
>
>
> Q2: Would it not be more beneficial to integrate CRSOM into CNN and compare this with a baseline CNN instead?
>
> R2: rRBF in this study is essential for implementing the knowledge infusion and cannot be replaced by another network like CNN. The reason is that the two-dimensional topological hidden layer of the rRBF offers us an intuitive understanding of the hidden representations, i.e., similar high-dimensional inputs are positioned close to each other, while dissimilar inputs are distanced on the map. This understanding allows humans to infuse their subjective knowledge by making topological arrangements on the map. In short, the rRBF offers direct translation for inputs's similarity on their original high-dimensional space into their relative distance on the low-dimensional representation space of the rRBF. In CNN, human cannot directly translate the difference of the input in their original high-dimensional space into the difference in their hidden representation. This explanation is added in lines 489-494.
>
> Integrating CRSOM into deeper networks like CNN will be more beneficial for increased performance. However, we found that the deeper layer will average out the infused human characteristics. Our immediate future study will address this trade-off, but at present, we consider our current model the best for infusing human knowledge into neural networks.
>
>
> Q3: the lack of clarity for legends
>
> R3: Thank you very much for pointing out this problem. We have now improved the clarity and caption of the figures.
>
>
> Q4: lack of clarity about standard CNNs used as comparisons in this study.
>
> R4: We have added the explanation about the structure of the CNN in this study in lines 376-377. All of the CNNs are composed of three convolutional layers, each one followed by a pooling layer, and subsequently two fully connected layers and finally a softmax layer.
>
>
> Q5: With only six participants, does this study have sufficient statistical power?
>
> R5: We admit that, at this point, the need for more participants is a problem in our experiment. However, as experiments involving humans are expensive and difficult to execute, at this stage, our primary goal is to establish a good framework for a new idea of infusing knowledge into neural networks. Although limited, this paper is a good start. Once we have a solid framework, we plan to execute experiments with many more participants, including online experiments with a variety of participants.
>
>
> Q6: Have ethical considerations been given to using human subjects in this experiment, including IRB approval?
>
> R6: Yes. All experiments are executed according to the ethical guidelines in the authors' institution. This explanation is added in lines 419-422.

---

> ### Author Response · Authors · 2024-11-25
> **Responses on the comments about the weakness of the proposed work (Nov. 25)**
>
> On the weakness of this paper mentioned by the reviewer.
>
> Comment1: If CRSOM identifies clusters that align with the arbitrary classified samples by the initializer, the authors' methodology could simply be considered a variant of SOM adapted to human prior knowledge (i.e., an application case).
>
> R1: We add our argument that CRSOM is substantially different from standard SOM. SOM is a non-supervised dimensionality reduction mechanism in which the data labels do not influence the topological arrangement. In contrast, CRSOM is influenced by the data labels, so it generates maps that consider the topological similarity of the inputs and their contexts (labels).
> This explanation is added in lines 152-171 and further mathematically elaborated in the Appendix in lines 600-632.
> Further, the proposed method generates a topological map and extracts an attention vector from the human initializer's initial arrangement, thus changing the distance metric that will be subsequently utilized for the network. These characteristics are absent in SOM.
>
>
> Comment 2:  Thus, even if knowledge infusion is feasible, further examination may be needed to confirm whether the information infused was indeed uniquely human (i.e., unobtainable by the model itself).
>
> R2: Thank you so much for your insightful comments. To address this point, we executed additional experiments using MNIST. While MNIST is a simple problem, this problem allows us to execute additional experiments to test the proposed idea against the variety of the quality of the infused knowledge. While humans subjectively perceive samples of MNIST, they can give a clear rationale for their perception. For example, most humans will consider digits "1" and "7" to be similar and digits "4" and "8" to be dissimilar. This will give variety on the subjectivity of the infused knowledge, including non-sensical knowledge infusion, i.e., organizing dissimilar digits close to each other. In the new experiments, we show that the quality of the infused knowledge subsequently influences the neural network, strengthening our argument that it is possible to build a neural network that learns from humans. This explanation is added in lines 319-323 and 451-480. For both OASIS and MNIST, we also executed experiments where the rRBFs are randomly initialized to show the superiority of human sensible initialization.

---

> > ### Comment · Reviewer_PDph · 2024-11-26
> >
> > I sincerely appreciate the authors’ thorough and thoughtful responses.
> > I have reviewed the updated version of this manuscript and observed significant improvements. In particular, the enhanced legends for the figures and the detailed descriptions of the experimental procedures are commendable. I deeply appreciate the authors’ efforts to understand my feedback and incorporate it into their work.
> >
> >
> > While I am considering revising my initial score, I would like to seek clarification on a few remaining concerns before making a decision:
> >
> > 1. The authors mentioned that CNNs cannot directly translate input differences in high-dimensional spaces into differences in hidden representations. However, in the field of metric learning, methodologies exist that address this issue regardless of the network architecture, including CNNs. The authors appear to emphasize that rRBF specifically tackles input differences in high-dimensional spaces directly. Nevertheless, even if this problem is addressed indirectly, as long as the performance is ensured, whether the approach is direct or indirect might be a secondary matter. If I have misunderstood this point, I apologize and would like to ask for the authors’ perspectives on this issue.
> >
> > 2. The authors indicated that an explanation regarding this concern was added between lines 259–264. However, the content in these lines seems unrelated to the matter at hand. Could the authors clarify this?
> >
> >
> > 3. For the OASIS and MNIST datasets, the authors compared rRBF initialized randomly with rRBF pre-trained using human knowledge. However, I question whether it is appropriate to compare a randomly initialized network with one pre-trained using human knowledge. I believe the comparison should be between a network pre-trained on features learned unsupervisedly by the neural network and one pre-trained using human knowledge. As the authors noted, most humans perceive the relationship between "1" and "7" as closer than that of "4" and "8." However, such observations can also emerge in representations unsupervisedly extracted by machine learning. There is evidence suggesting significant overlap between features or manifolds learned through machine learning and those perceived by humans. What are the authors’ thoughts on this?

---

> ### Author Response · Authors · 2024-11-26
> **Response to Official Comment by Reviewer PDph**
>
> I sincerely thank you very much for checking our modified paper thoroughly and for acknowledging our efforts to improve it..
>
> 1. There is a misunderstanding here. We did not mean that CNN could not translate, but we tried to argue that humans could not make the translation. In our proposal, it is essential for humans to translate the difference between samples in their original high-dimensional space and their difference in the representation space during the knowledge infusion process. While in rRBF the difference can be directly translated into the distance on the topological map, humans cannot, at least intuitively, understand the difference in the internal representations of CNN. Hence, it is essential to have a neural network that accommodates this intuitive translation. This point is explained in lines 489-494 in the paper (I apologize for my previous mistake of pointing out the lines for this explanation).
>
> 2. Thank you very much for this insightful discussion. This paper proposes a way to infuse human knowledge into a neural network. We do not claim that this always leads to a better performance. We tried to argue that sensible knowledge will help the neural network to learn and will be inherited after the learning process is terminated. Here, we assume sensible human knowledge infusion intrinsically includes good knowledge that benefits the neural network. However, as the reviewer mentioned, this knowledge does not have to always come from humans. Any sensible knowledge infusion from, for example, Autoencoder will also do the job. The source of the knowledge will not be distinguishable by the neural network. However, the point is that our proposal allows humans to make novel interactions with neural networks that could not have been possible previously. It is not our intention to argue that human-initialized neural networks are always better than neural networks initialized by other means. For this reason, we do not compare our network with autoencoders-initialized networks.
>
> Random initialization in the experiments was meant to illustrate the absence of sensible knowledge. The experiments in Fig. 5 and Fig. 8 show that the rRBF benefits from sensible initialization compared to "no knowledge" initialization. However, Fig. 8 shows that random initialization is still better than non-sensical initialization. This fact strengthens our argument that building a neural network that inherits the initializer's knowledge is possible.
>
> Once again, I thank you very much for this insightful discussion.

---

> > ### Comment · Reviewer_PDph · 2024-12-03
> >
> > Thank you for the authors' diligent responses. Their replies have addressed some of my questions and clarified certain points. While this paper still suffers from limitations such as insufficient statistical power and the relative incompleteness of the initial version, it also has strengths, including solid motivation and a relatively sound methodology. Considering the high costs associated with human behavioral experiments, it is important to acknowledge the contributions of studies based on small samples for advancing the field of human-in-the-loop research. Despite its merits, it remains necessary to assess whether this paper has reached the level appropriate for presentation at ICLR. I am still contemplating the possibility of updating the rating for this paper and will make a decision after careful consideration.

---

> > > ### Author Response · Authors · 2024-12-03
> > > **response to reviwer PDph**
> > >
> > > Thank you very much for your comment and for acknowledging the motivation and viability of the proposed HITL system.
> > > We admit that many weaknesses still need to be addressed. We hope this paper will establish a solid framework for advancing this idea further, especially for experiments involving many more participants, which will allow us to execute solid statistical analysis.
> > >
> > > Thank you again for your thoughtful comments that helped us improve this paper.

---

### Official Review · Reviewer_PVd4 · 2024-11-03

**Soundness:** 2
**Presentation:** 1
**Contribution:** 2
**Rating:** 3
**Confidence:** 3

**Summary:**

This paper presents a novel method for infusing human knowledge into neural networks by constructing a Restricted Radial Basis Function (rRBF) network, which incorporates human knowledge, experience, and preferences into the initialization and retraining phases of the network. The paper demonstrates the application of this method in Alzheimer's disease detection and compares its performance to standard neural networks, with experimental results validating its feasibility. This research provides an innovative approach for human participation in the AI learning process, opening up new possibilities for human-AI interaction.

**Strengths:**

Innovative method：

The paper presents an innovative approach for directly infusing human knowledge into neural networks through a Restricted Radial Basis Function (rRBF) model, expanding traditional human-in-the-loop (HITL) methods.
Application to Alzheimer’s detection provides a meaningful, high-impact example of embedding human insights into healthcare AI.
Quality:

**Weaknesses:**

1. **Scope of Experiments**:

   The experiments focus on Alzheimer's disease detection using MRI data. However, the study would benefit from broader experimental validation across other tasks or datasets to assess the generalizability of the HITL rRBF approach. Applying this framework to different domains, especially those where data interpretation is less subjective, could help confirm the flexibility and robustness of the method.

2. **Baseline Comparisons**:

   Although the paper includes comparisons to non-HITL models, such as standard CNNs, it could be strengthened by including additional HITL benchmarks. For instance, other recent HITL approaches, or self-organizing map-based methods, could serve as complementary baselines. This would provide a more comprehensive assessment of how the proposed model stands in comparison to existing HITL techniques.

3. **Depth of Analysis on Human Knowledge Infusion**:

   While the paper demonstrates that human initialization improves model performance, the impact of specific types of human input (e.g., different expertise levels or subjective biases) is not explored in depth. Understanding how variations in human knowledge influence the model could clarify the boundaries and limitations of the infusion method, especially for practical deployment in diverse real-world applications.

4. **Reproducibility and Scalability**:

   The paper states that the rRBF method relies on human organization of inputs, which raises questions about scalability for larger datasets. Addressing how the method could be adapted to datasets where human organization is not feasible, or discussing a hybrid approach combining human knowledge with automated processes, could enhance the method’s practicality.

**Questions:**

no

---

> ### Author Response · Authors · 2024-11-25
> **Responses on the comments about the weakness of the proposed work (Nov. 25)**
>
> Thank you very much for your insightful comments on the weakness of our proposed model.
>
> Comments: Application to different data, scalability, impact on different expertise levels.
>
> Response: Thank you for raising these critical points.
> Ideally, the experiments should be run using a pool of medical doctors with different expertise levels. Unfortunately, as experiments using doctors are difficult to execute, we cannot perform them at this point. At this stage, we want to establish a solid method that will allow us to test our proposed HITL mechanism further for real-world problems. To complement this weakness, we ran tests against MNIST that, to some extent, allowed us to observe the effect of the different levels of expertise. While MNIST is a simple problem, this problem allows us to execute additional experiments to test the proposed idea against the variety of the quality of the infused knowledge. While humans subjectively perceive samples of MNIST, they can give a clear rationale for their perception. For example, most humans will consider digits "1" and "7" to be similar and digits "4" and "8" to be dissimilar. This will give variety on the subjectivity of the infused knowledge, including non-sensical knowledge infusion, i.e., organizing dissimilar digits close to each other. In the new experiments, we show that the quality of the infused knowledge subsequently influences the neural network, strengthening our argument that it is possible to build a neural network that learns from humans. This explanation is added in lines 321-323 and 403-422. For both OASIS and MNIST, we also executed experiments where the rRBFs were randomly initialized to show the superiority of human sensible initialization. This additional experiment also, to some extent, demonstrates that the proposed method is scalable to the data size in that humans only need a small part of the data to organize before the rRBF takes over.

---

### Official Review · Reviewer_UQ9K · 2024-11-03

**Soundness:** 3
**Presentation:** 3
**Contribution:** 3
**Rating:** 5
**Confidence:** 4

**Summary:**

The paper proposes a method of infusing human knowledge into neural networks with two-dimensional topological hidden representations called restricted Radial Basis Function Networks. The method has been tested in Alzheimer's image data.

**Strengths:**

- The idea is very good and novel and would be a good contribution to the community.
- The paper is well written, the idea is clear and the presentation is good.

**Weaknesses:**

- Poor evaluation with limited experiments and even more limited comparisons. The proposed method is validated only in one medical dataset. I would suggest to test it against other datasets too. Regarding the comparisons I understand that this is more difficult but you need to figure out a good ablation study at least.
- The method is applied only on one neural network which is considered not black-box. I would highly recommend to apply it in other regular networks or at least try to generalize it.

**Questions:**

Why do you need the rRBF and you can't just do the experiments in a regular NN?
How does $\Lambda$ if instead of having the human input in the initialization you have it after the training? since you mentioned in the beginning of the paper that the infusion can be executed in two different stages of the neural network training.

The figures should be self-contained with better descriptions and with higher quality of the figure. For example Figure 7 looks unprofessional and not fit for this venue.

---

> ### Author Response · Authors · 2024-11-14
> **response to reviewer's questions (Nov. 25)**
>
> Thank you very much for your constructive questions.
> We have now revised our paper.
>
> As for your questions, our point-to-point responses are as follows.
>
> Q1: Why do you need the rRBF and can't just run the experiments in a regular NN.
> R1: We need to do the knowledge infusion in rRBF because the two-dimensional topological hidden layer of the rRBF offers us an intuitive understanding of the hidden representations, i.e., similar high-dimensional inputs are positioned close to each other, while dissimilar inputs are distanced on the map. This understanding allows humans to infuse their subjective knowledge by making topological arrangements on the map. In short, the rRBF offers direct translation for inputs's similarity on their original high-dimensional space into their relative distance on the low-dimensional representation space of the rRBF. These arrangments cannot be done on the regular NN, for example, DNN, where the hidden representations are not interpretable by humans.  This explanation is added in lines 489-494.
>
>
> Q2: How $\Lambda$ does if instead of having the human input in the initialization you have it after the training?
>
> R2: We already showed the difference between  $\Lambda$  before and after the training, as the heatmaps in Fig. 3 and Fig. 4. The heatmaps in Fig. 3 are the visualization of the initial $\Lambda$, while the heatmaps in Fig. 4 are the visualizations of the "revised" $\Lambda$  after the retraining.
>
>
> Q3: Figure should be self contained.
>
> R3: All the figures are corrected.

---

> ### Author Response · Authors · 2024-11-25
> **Responses on the comments about the weakness of the proposed work (Nov. 25)**
>
> On the reviewer's comment on poor analysis and limited experiments.
> Thank you very much for your constructive criticism.
> To address this weakness, we added experiments against MNIST. While MNIST is a simple problem, this problem allows us to execute additional experiments to test the proposed idea against the variety of the quality of the infused knowledge. While humans subjectively perceive samples of MNIST, they can give a clear rationale for their perception. For example, most humans will consider digits "1" and "7" to be similar and digits "4" and "8" to be dissimilar. This will give variety on the subjectivity of the infused knowledge, including non-sensical knowledge infusion, i.e., organizing dissimilar digits close to each other.  In the new experiments, we show that the quality of the infused knowledge subsequently influences the neural network, strengthening our argument that it is possible to build a neural network that learns from humans. The explanation is added in lines 321-323 and 403-422.

---

### Comment · Area_Chair_uacz · 2024-11-25
**Please engage in the discussion**

Dear all,

Many thanks to the reviewers for their constructive reviews and the authors for their detailed responses.

Please use the next ~2 days to discuss any remaining queries as the discussion period is about to close.

Thank you.

Regards,

AC

---

### Author Response · Authors · 2024-12-02
**Final general comments to all reviewers**

I want to thank you very much for reviewing our paper and providing insightful comments.
I admit that we cannot sufficiently address some of the reviewer's concerns. However, after the revision, I believe the paper's technical quality and readability have improved. We have also added some new experiments to assess the effects of the quality of the infused knowledge on the neural network's performance.

I would be grateful if the reviewers could re-review and re-assess the paper.

---

### Meta-Review · Area_Chair_uacz · 2024-12-18

**Metareview:**

This paper introduces a novel method for integrating human knowledge into neural networks by constructing a Restricted Radial Basis Function (rRBF) network. This network incorporates human knowledge, experience, and preferences during its initialisation and retraining phases. The paper demonstrates the application of this method in Alzheimer’s disease detection and compares its performance to conventional neural networks, with experimental results validating its feasibility. This research presents an innovative approach to human involvement in the AI learning process, opening up new possibilities for human-AI interaction.

The experiments focus solely on Alzheimer’s disease detection using MRI data, which limits the generalisability of the approach. Broader experimental validation across other tasks or datasets is essential to assess the model’s applicability in diverse domains, particularly those where data interpretation is less subjective. Applying this framework to different domains would enhance the method’s generalisability and robustness.

While the paper shows that human initialisation enhances model performance, it doesn’t explore the specific impact of different types of human input, such as varying expertise levels or subjective biases. Understanding how variations in human knowledge influence the model could clarify its boundaries and limitations, particularly for practical deployment in diverse real-world applications.

Lastly, given the above, the technical novelty of the paper is rather limited considering ICLR's remit, and other venues such as MICCAI might be more suitable for the work presented here. I would like to encourage the authors to revise their submission and submit it to another venue in the near future.

**Additional Comments On Reviewer Discussion:**

Reviewers have been well-aligned in their opinions about this paper. The authors responded to some of the queries but the key limitations identified by the reviewers are part of the paper's backbone therefore hard to change during the rebuttal, e.g. human input, technical novelty, evidence of generalisability of the methods, etc.

I acknowledge, though, that the authors tried to meet the reviewers halfway by adding some results on MNIST

---

### Decision · Program_Chairs · 2025-01-22

Reject